# Systematic Review of Photodynamic Therapy in Gliomas

**DOI:** 10.3390/cancers15153918

**Published:** 2023-08-01

**Authors:** Tiffaney Hsia, Julia L. Small, Anudeep Yekula, Syeda M. Batool, Ana K. Escobedo, Emil Ekanayake, Dong Gil You, Hakho Lee, Bob S. Carter, Leonora Balaj

**Affiliations:** 1Department of Neurosurgery, Massachusetts General Hospital, Boston, MA 02114, USA; 2Chan Medical School, University of Massachusetts, Worcester, MA 01605, USA; 3Department of Neurosurgery, University of Minnesota, Minneapolis, MN 554414, USA; 4Center for Systems Biology, Massachusetts General Hospital Research Institute, Boston, MA 02114, USA; 5Department of Radiology, Massachusetts General Hospital, Boston, MA 02114, USA; 6Harvard Medical School, Boston, MA 02215, USA

**Keywords:** glioblastoma, photomedicine, photodynamic therapy, photosensitizer

## Abstract

**Simple Summary:**

Malignant gliomas are of the deadliest, most hard-to-treat cancers, given their various characteristics of aggression and infiltration as well as the location of growth. Photodynamic therapy (PDT) is a promising avenue for localized cancer therapy. A therapeutic effect is achieved by systemic drug delivery followed by localized wavelength-specific illumination. This review extensively explores the use of photosensitizers in gliomas, from their first use to the present, and details their mechanisms, and pre-clinical and clinical findings. Further discussion is provided on its limitations and future directions.

**Abstract:**

Over the last 20 years, gliomas have made up over 89% of malignant CNS tumor cases in the American population (NIH SEER). Within this, glioblastoma is the most common subtype, comprising 57% of all glioma cases. Being highly aggressive, this deadly disease is known for its high genetic and phenotypic heterogeneity, rendering a complicated disease course. The current standard of care consists of maximally safe tumor resection concurrent with chemoradiotherapy. However, despite advances in technology and therapeutic modalities, rates of disease recurrence are still high and survivability remains low. Given the delicate nature of the tumor location, remaining margins following resection often initiate disease recurrence. Photodynamic therapy (PDT) is a therapeutic modality that, following the administration of a non-toxic photosensitizer, induces tumor-specific anti-cancer effects after localized, wavelength-specific illumination. Its effect against malignant glioma has been studied extensively over the last 30 years, in pre-clinical and clinical trials. Here, we provide a comprehensive review of the three generations of photosensitizers alongside their mechanisms of action, limitations, and future directions.

## 1. Introduction

Glioma is the most common malignant primary central nervous system tumor type and consists of several subtypes, including glioblastoma (GBM) which make up 57.3% of all gliomas [1]. An extremely aggressive subtype, GBM is characterized by its high infiltrative and angiogenic attributes [2]. The current standard of care for gliomas involves maximal surgical resection followed by radiation therapy with concurrent and adjuvant temozolomide (TMZ). For GBM, in particular, this regimen is known as the Stupp protocol and yields a median survival of 14.6 months, a median progression free survival (PFS) of 6.9 months, and a two year overall survival (OS) rate of 26.5% [3,4,5]. Often, treatment failure can be attributed to factors including inter- and intratumoral heterogeneity, reacquisition of stemness in glioblastoma stem cells, the evolution of therapy-resistant clonal subpopulations, the tumor-promoting microenvironment, multiple drug efflux mechanisms, metabolic adaptations, and enhanced repair of drug-induced DNA damage [6,7]. As such, disease recurrence is nearly inevitable in patients with high-grade gliomas. Despite advances in treatment strategies, there are currently no standard therapeutic options for recurrent glioma and recurrent GBM (rGBM) management. Over the past two decades, therapeutic strategies have been explored but with minimal success. Trials using monotherapeutic and combinatorial drugs have explored the use of nitrosoureas, antiangiogenics, and EGFR inhibitors to treat rGBM [8,9]. Of these therapies, trials involving the use of bevacizumab in TMZ-pretreated patients have shown promising results [10]. However, there have been no significant differences in patient survival across the large host of regimens, with many trials often resulting in an increased risk of toxicity and adverse events [9,11,12]. With no standard treatments, patients are treated conditionally according to clinical characteristics and conventional prognostic factors.

Over the last (almost) 50 years, the utility of photosensitizers (PS) in the context of fluorescence-guided surgery (FGS) and photodynamic therapy (PDT) in glioma have advanced with rapidly growing momentum [13,14,15,16]. PDT is a two-stage treatment that combines light energy with a drug (photosensitizer, PS) designed to destroy cancerous and precancerous cells after light activation (Figure 1). PDT relies on a tumor-selective, otherwise inert, PS molecule that is administered locally or systemically. Whereas standard radio- and chemotherapies act non-specifically, PDT selectively targets tumor tissue due to the preferential accumulation of the drug in malignant tissue. When excited by light of a particular wavelength, PSs will absorb the energy, convert into an intermediary byproduct, and undergo intersystem crossing [17]. This phenomenon leads to the buildup of tumoricidal molecules, such as reactive oxidative species (ROS), resulting in localized destruction of the tumor [18,19,20,21]. Given the tumor-specific and tumor-targeting nature of PSs, the use of PDT becomes extremely attractive for treating both GBM and rGBM, given its high cytotoxicity, minimal normal tissue toxicity and systemic effects, and minimized risk of local recurrence [19,22]. Further, in comparison to standard chemo- and radiotherapies, PDT is performed during surgery and often results in fewer side effects, providing a significant advantage to patient well-being [23].

In this review, we explore the key advancements of photosensitizers and photodynamic therapy, as well as their advantages and pitfalls, focusing on future therapeutic perspectives for the management of gliomas.

## 2. Photodynamic-Therapy-Mediated Tumoricidal Effect

### 2.1. PDT Induces Cell Death following PS Uptake, Accumulation, and Activation

Unlike temozolomide (TMZ) which acts by destabilizing DNA, the tumoricidal effect of photodynamic therapy (PDT) manifests by causing oxidative damage to cell membranes, proteins, and organelles. This triggers a combination of necrosis and apoptosis, as well as immunogenic and autophagy pathways [18,19,20]. It is well-known that standard radiation therapy for malignant glioma can cause radiation necrosis of the treated tissue. This may induce headache, vomiting, loss of consciousness, and hemiplegia in patients, but can be relieved by surgical removal of necrotic areas [24]. Therefore, apoptosis and autophagy are considered the preferred cell death mechanisms for glioma treatment modalities. Distinct types of PDT-induced cell death vary depending on (i) PS subcellular localization, pharmacokinetics, and chosen drug dose; (ii) light dosimetry; (iii) tissue oxygenation status; and (iv) tumor-subtype specific properties [25,26,27,28]. PSs that localize in the mitochondria have been shown to induce apoptosis, whereas PSs that localize in lysosomes and plasma membranes generally demonstrate a necrotic cellular response [29]. However, these different cell death types are not mutually exclusive and have the potential to co-occur or shift from one survival pathway to another by altering just one variable in the treatment schema [24,30]. In addition to the above-mentioned modalities, PDT has also been shown to trigger ferroptosis-like cell death in glioma [31]. In fact, there exists a synergistic effect of the intrinsically regulated cell death process ferroptosis and PDT, with the two processes yielding elevated reactive oxygen species (ROS) for increased anti-cancer effect [32].

### 2.2. PDT Controls Glioma Stem Cell (GSC) Processes

In addition to their effects on the glioma cells, PDT can also curtail the growth of GSCs and induce their death [33,34]. GSCs are multipotent cells with tumorigenic capability [35], driving tumor regeneration and disease progression, and are predominantly involved in the recurrence of GBM [36]. Studies have shown that long-term TMZ exposure increases GSC subpopulations [35]. Recent evidence also suggests that in vitro GSCs accumulate protoporphyrin IX (PpIX), a downstream product of the second-generation PS 5-aminolevulinic acid (5-ALA), to therapeutic levels in a dose- and time-dependent manner, inducing cell death [33]. More recently, Fisher et al. described a doubling of median OS in rats bearing GSC-30 tumors following low-dose ALA-PDT combined with lapatinib, an EGFR inhibitor, as compared to rats with conventional U87 human glioma cell line tumors [37].

### 2.3. PDT Modulates Neurovasculature: Disruption of the Blood–Brain Barrier (BBB) and Destruction of Tumor Vasculature

PDT mediates the breakdown of the BBB through several avenues including increasing gaps between tight junctions, microtubule depolarization, and imbalanced endothelial regulation of vascular relaxation [38]. Subsequent BBB breakdown facilitates the diffusion of PSs into the brain and tumoral area. It is little known if PSs, such as 5-ALA, are proxies for the tumor tissue or just a manifestation of BBB disruption. In a proof-of-concept report, Madsen et al. showed exogenous macrophage migration into the brain of non-tumor bearing mice following PDT-induced BBB disruption [39]. Later reports have also optimized the use of certain PSs (particularly, 5-ALA) and PDT for BBB disruption with positive results [38]. The breakdown of the BBB can result in increased accumulation of PS, over nonspecific therapeutic delivery vehicles, in the tumor tissue, compounding the antitumor effect.

Similarly, PDT also bears an effect on tumor vasculature. Angiogenesis is an important hallmark of cancer, contributing to tumor resilience and aggressive growth [40]. Intriguingly, there is evidence of microvascular constriction, collapse, and thrombus formation following PDT, ultimately delaying or inhibiting tumor growth [41]. Characterization of tumor morphology following PDT has revealed lapsed sinusoids, intumescent endothelial cells within tumor capillaries, luminal occlusion, and thrombosis [42]. Combined, these two effects on the neurovascular system compound the therapeutic effects of PDT on malignant gliomas.

### 2.4. PDT Stimlates Anti-Tumor Immunity

Photodynamic therapy has been implicated in macrophage, immune cell, and T cell recruitment and enhanced anti-tumor immunity [20,43,44]. Emerging studies have also described an increased migration of antigen-presenting cells, such as macrophages [39] and dendritic cells [45], and cytokines [46] to brain tissue treated with PDT. This immunological effect is mediated by an upregulation of damage-associated molecular patterns (DAMPs), such as heat shock proteins (HSPs) [47,48], surface calreticulin (CRT), secreted adenosine triphosphate (ATP), and high-mobility group box 1 protein (HMGB1) [49]. DAMPs are released by damaged or dying cells to stimulate vascular permeability and production of proinflammatory cytokines, thereby mediating leukocyte migration to the site of tissue damage. The subsequent increase in local inflammation further triggers immune cell recruitment. Through both in vitro and in vivo studies, there is significant evidence of PDT treatment-induced upregulation of HSP70 surface expression [47,50,51]. In rat models, accumulation of CD8+ T cells and macrophages/microglia were seen in conjunction with HSP70 upregulation following nanoparticle-based PDT [48]. Studies employing mixtures of photosensitizers have also prompted phagocytosis of glioma cells via bone marrow dendritic cells (BMDCs), via BMDC maturation and production of IL-6 in a cell-ratio-dependent manner [49]. Genes for immune protein markers IL-6 and IL-6R have also been found to be upregulated alongside ROS-inducible genes after PDT [52]. Over the last few years, in vitro studies have moved towards investigating non-porphyrin, third generation PSs, with many utilizing cross-linked polymers and nanoplatforms for targeted delivery to enhance PDT-induced immunological tumoricidal effects [53,54,55,56,57,58,59,60].

## 3. Photosensitizers

Clinically efficacious photosensitizers (PSs) are constrained by several properties, namely (i) ability to penetrate the BBB, (ii) selective accumulation in malignant tissue, and (iii) photoactivity at long light wavelengths for deep tissue penetration [61]. PSs are categorized into three generations based on molecular properties (Table 1 [19,62,63,64,65,66,67,68,69,70,71,72]): first-generation PSs are naturally occurring porphyrins, second-generation PSs comprise more chemically pure and tumor-selective compounds, and third-generation PSs broadly cover engineered nanoplatforms, gene-engineered, and carrier-bound systems that further enhance tumor-selective cytotoxicity [19,73,74].

### 3.1. First Generation: Naturally Occurring Porphyrins

First-generation PSs include hematoporphyrin (HP) and the purified derivatives of porfimer sodium (Photofrin), hematoporphyrin derivative (HpD, Photofrin I), and dihematoporphyrin ether (HPE, Photofrin II) [29]. The use of hematoporphyrin derivatives in malignancies was reported as early as 1960–1961 [75,76], with their use as a PS in glioma PDT ensuing a few decades later. The notable tumor-localizing characteristic of PSs was identified twenty years later [77] and shown to be attained following alkaline hydrolysis of the drug. Interestingly, these porphyrin-based PSs bear extreme structural resemblance to that of heme. Yet, unlike heme, these structures lack Fe ion coordination, thereby preventing downstream oxidative conversion and catabolism, resulting in cellular accumulation and subsequent PDT-induced toxicity [78,79].

Since the 1980s, first-generation photosensitizers have been clinically tested in numerous safety and feasibility studies for malignant glioma treatment (Table 2 [13,65,80,81,82,83,84,85,86,87,88,89,90,91,92,93,94,95,96,97,98]). In comparison to the current standard of care and results from other PS studies, HP and its derivatives have demonstrated modest survival outcomes. From 1980 to 1990, several groups from Italy [80,99], Australia [81,100], and the United States [13] reported successful use of PDT with HpD for treatment of glioma in small cohorts of patients. Across these studies, low doses of HpD (1–5 mg/kg) were administered via intratumoral, intra-arterial, or intravenous injections prior to photo-illumination via a variety of light delivery and optical energies [101].

To further improve patient survival, studies combining PDT with standard therapeutic modalities such as radiation were implemented [85]. In 1988, a Phase I/II clinical trial combined PDT with single-shot ionizing radiation and/or conventional radiotherapy to assess combined treatment efficacy across three populations of patients: primary glioma, single-recurrence GBM, and multiple rGBM [86]. HPD was administered either through the internal carotid artery or directly into the tumor bed. Using low-to-moderate light doses (60–200 J/cm^2^), subsequent PDT was performed. In general, outcomes for recurrent disease patients were unimpressive, with patient progression free survival (PFS) ranging from 2–15 months. Furthermore, patients who had previously failed conventional radio- or chemotherapy did not benefit from the PDT–RT combination. In a larger follow-up study with 50 GBM patients, the authors increased the light dose (200 J/cm^2^ to 250 J/cm^2^) and PS concentration (1 mg/kg to 2.5 mg/kg), noting improved PFS for both primary GBM (*n* = 11) and rGBM (*n* = 39) at 13 months and 7 months, respectively [102,103].

Interestingly, a positive association has been shown between increased light doses and improved outcomes across several groups ranging from initial to current studies. A retrospective study on HpD-treated patients (*n* = 136) treated from 1986 to 2000 noted significantly improved prognosis (hazard ratio = 0.502) in primary tumor cases treated with doses of 230 J/cm^2^ or higher [96]. In recurrent tumor cases, however, increased light doses effectuated only a slight improvement in the hazard ratio (HR = 0.747). While it may be argued that these improvements are confounded by advances in technology, namely increased laser light stability across different laser sources, other concurrent studies on similar patient populations (age, baseline Karnofsky score) [93] identified similar improvements in median OS as laser light dosage increased (<1200 J: 39 weeks, >1200 J: 52 weeks).

As with first- and second-generation PDT in other systemic cancers [104], other light sources such as LEDs have also been explored. A study comparing laser- and LED-based PDT demonstrated similar levels of tissue toxicity at equivalent light doses across several types of tumors, including GBM [65]. Given the broader emission spectrum and increased major emission wavelength of LED light, however, the study concluded that PSs with higher absorption peaks, such as benzoporphyrin derivatives (BPD, a second-generation PS), were more suitable for LED-based PDT.

While first-generation PSs have demonstrated successful PDT effects in glioma [105,106], the chemical properties of these compounds limit their efficacy as ideal PDT candidates. Firstly, this generation of PSs is limited, not only by their low therapeutic efficacy, but these drugs bear a low singlet oxygen quantum yield [107]. Furthermore, the PDT response using first-generation PSs has been inconsistent across different glioma cell lines [108]. When compared to second-generation PSs, there are notable differences in efficacy, as described by relatively higher non-specific PS accumulation in normal brain tissue, longer extended illumination times, and increased elimination half-life. While porfimer sodium has a higher absorption spectrum than HpD, both HpD and porfimer sodium have extended half-lives and renal clearance time [109], with porfimer sodium persisting in circulation over 2 months, increasing the risk of unwanted photo-related toxicities [19,73,110]. The structural nature and size of these naturally occurring porphyrins, large tetrapyrrole macrocycles connected via methine bridges, respectively, exhibit aggregation in water and, additionally, complicate delivery across the blood–brain barrier. Combined, first-generation PSs exhibited several limiting characteristics that hindered their utility for glioma care. As such, the development of second-generation photosensitizers was warranted to expand the efficacy of glioma photodynamic therapy.

### 3.2. Second Generation: Increased Singlet Oxygen Potency

In contrast to first-generation PSs, second-generation therapeutics showed improved purity, more efficient ROS production, and enhanced tumor selectivity with limited adverse effects. Second-generation PSs mostly consist of porphyrin or chlorin-based structures and precursors. Those include 5-aminolevulinic acid (5-ALA; Gliolan), Talaporfin sodium (mono-L-aspartyl chlorin e6, NPe6, TS; Laserphyrin), boronated porphyrins (BOPP), Temoporfin (m-THPC, Foscan and Foslip), and benzoporphyrin derivatives (BPD; Verteporfin) [63,73]. These newer second-generation PSs bear phototoxic properties at a longer light wavelength than first-generation PSs and can be excited at lower energies (down to 20 J/cm^2^), yielding a greater potential to target deeper tumor tissue [111]. Over the last three decades, clinical trials have assessed second-generation PSs for their treatment of gliomas (Table 3 [22,62,65,72,111,112,113,114,115,116,117,118,119,120,121,122,123,124,125,126,127,128]). Below, we discuss two of the most common second-generation photosensitizers studied under in vitro, in vivo, and clinical conditions. However, extensive evaluation of other second-generation PSs such as metallo-phthalocyanines have also ensued in parallel for both pediatric and adult brain tumors [129,130,131,132].

To this day, 5-ALA is one of the most heavily explored and utilized PSs, and has been established as a valuable tool for both real-time intraoperative visualization (fluorescence guided surgery, FGS) for malignant glioma resection and as a tumor-selective PS for PDT. Administered orally, 5-ALA is an endogenous compound that is metabolized via the heme pathway. Unlike first-generation photosensitizers, 5-ALA is a molecular precursor for heme, therefore relying on a different mechanism of action for therapeutic activation. In tumor cells, suppression of membrane transport proteins and dysregulation of the Ras/MEK and FECH/heme oxygenase pathways result in the buildup of protoporphyrin IX (PpIX), particularly following exogenous dosage [136,137,138,139]. The subsequent buildup of PpIX exerts fluorescent and phototoxic properties on the tumor cell upon excitation with blue (405 nm, FGS) or red (635 nm, PDT) light [140]. 5-ALA offers advantageous clinical benefits due to its high tumor selectivity, rapid renal clearance, and limited adverse effect on normal brain tissue [61]. Several clinical studies have demonstrated high drug efficacy and improved survival in malignant glioma patients. Particularly, a study comparing the effects of 5-ALA PDT on median survival in patients with GBM distinguished an improvement of over 2-fold (62.9 weeks vs. 20.6 weeks) following intraoperative cavitary therapy [119].

Compared to 5-ALA, talaporfin sodium (TS) is delivered intravenously and does not exert tumor selectivity through metabolic pathways. Rather, it circulates in the blood, conjugated to albumin, and accumulates selectively in tumor cells via lysosome endocytosis, relying on increased vascular permeability at the blood–tumor barrier. Consequently, blood vessel endothelium, blood that has accumulated in the resection cavity, and non-glioma vascularized CNS neoplasms have been observed to exhibit marked fluorescence, a characteristic that negatively impacts its performance as a tumor-selective PS [72,141]. Schimizu et al. have demonstrated the feasibility of TS-based FGS for GBM, demonstrating a specificity of 80% and sensitivity of 71% with strong fluorescence, was noted in both newly diagnosed and rGBM. Recent studies have reported the feasibility of TS-based PDT, via mitochondrial-mediated apoptosis and necroptosis at low TS doses and low laser irradiation [24,142,143]. However, as TS doses increased, evidence suggests that the dominant modality of cell death shifts from apoptosis to necrosis [30]. The mechanisms underlying these variations in mechanisms and the extent of cell death in different glioma cells require further exploration to optimize clinical application [142].

Current PDT clinical trials around the world continue to assess the efficacy of second-generation photosensitizers for malignant gliomas (Table 4). In recent years, in particular, studies have begun investigating the utility of both TS and 5-ALA in pediatric brain tumor patients, with age enrollment criteria ranging from 3 to 20 years old (UMIN000030883: 6 years to 20 years; NCT04738162: 3 years to 17 years). While the listed studies have yet to report findings, study outcomes will encompass drug safety and tolerability, OS, and PFS.

#### Combining Second-Generation PDT with Standard Therapies

In addition to single-drug PDT, studies have investigated the synergistic effect of second-generation PSs with standard care modalities. A notable study, performed in 2012, investigated the efficacy of 5-ALA PDT alone, against standard maximum safe resection (MSR) and alongside intraoperative radiotherapy (IORT) to assess patient PFS across four different cohorts: (i) MSR only, (ii) MSR + IORT, (iii) MSR + PDT, (iv) MSR + PDT + IORT (Table 3, Lyons et al. (2012)) [119]. Aside from the significant effect of PDT on median survival, adding IORT to the treatment regimen (iii vs. iv) increased PFS from 39.7 weeks to 79 weeks.

Studies combining PDT with standard chemotherapies have been performed as well. Two separate groups utilizing different PS-based PDT protocols found that TMZ concomitant therapy potentiates PDT-induced apoptotic cell death in vitro [144,145]. The combination of TS and TMZ delivery increased intracellular concentrations of PS and upregulation of ROS production following cotreatment [145]. Additionally, in vitro study has shown that second-generation guided PDT and TMZ act synergistically to decrease glioma migration and invasiveness in glioma cells by downregulating the protein NHE1, preventing escape from TMZ-mediated toxicity [144,146]. Further building on this combined efficacy, in vivo rat models have found that PDT increases TMZ tissue concentrations and combinatorial treatment decreases tumor volume and prolongs survival [147]. In addition to co-delivery with the standard TMZ, studies have investigated the use of PDT with anti-angiogenic drugs. A combination of PDT with bevacizumab has demonstrated increased median survival time in glioma-bearing rats, as compared to PDT or bevacizumab alone [148].

Additional studies have demonstrated positive effects of PDT as a standalone mediator of cancer immunotherapy [149], and have also identified the synergistic effects of PDT in combination with immunotherapies, such as PD-L1 checkpoint blockade therapy. In a study investigating chlorin e6-mediated PDT in combination with PD-L1 checkpoint blockade therapy, glioma orthotopic mice were found to have significantly improved survival as compared to both naive and monotherapy conditions (anti-PD-L1 checkpoint blockade or PDT alone) [150]. In several other types of orthotopic cancer models, including lung [151,152,153], colorectal [153,154] and breast cancers [153,155], melanoma [153,156], and renal adenocarcinoma [157], combination of photodynamic therapy with anti-PD-L1 therapies have enhanced antitumor immunity and subsequent survival. The immunosuppressive tumor microenvironment common to gliomas combined with enhanced local inflammatory response, as a result of PDT, may affect subsequent immune cell localization and infiltration into the tumor. This, in turn, may result in some level of immune resistance following PDT. While detailed mechanisms of action continue to be clarified, studies have found evidence of PDT-induced reduction of PD-L1 expression on glioblastoma tumor cells, which will serve to magnify the anti-PD-L1 therapeutic effect [158]. Further study on the co-effects of PDT and chemo- or immune-therapies have been additionally investigated with the development of third-generation photosensitizers.

### 3.3. Third-Generation PS: Increased Tumor Selectivity

Beginning in the early 2000s, in vitro PDT studies for GBM shifted towards developing and optimizing nanomedicine delivery systems [159]. Compared to first- and second-generation PSs, third-generation PSs tend to have increased local specificity [159], enhanced cellular PS internalization, and improved PS retention. Broadly, third-generation PSs are composed of a broad spectrum of delivery vehicles that have been expounded on, including polymer- or lipid-based carriers such as liposomes [48,160], organometallic complexes [161,162,163,164], albumin- or antibody-conjugated nanospheres and nanocapsules [150,165,166], micelles [167,168], dendrimers, nanocrystals, and nanogold [169]. Most commonly, many groups have attempted to encapsulate a clinically-approved PS, such as BPD [170,171], m-THPC [172], chloro-aluminum phthalocyanine (AlClPc) [173,174], or indocyanine green [48], within biocompatible nanoparticles and nanoemulsions for more controlled drug delivery and release. In recent years, studies inducing PDT via other anti-tumor drugs [161], chemotherapy combination [175], and upconversion [176] modalities have developed as well.

Upconversion nanoparticles (UCNP) are nanoparticles (NP) that are doped with heavy metals which, when excited, upconvert wavelengths (such as near-infrared) to produce shorter emission wavelengths on the visible or UV scale [177]. Functionalized UCNPs, produced following conjugation with PSs, have been found to induce photodynamic therapeutic effects in GBM via near-infrared triggering [176,178]. Groups have also focused on developing synthetic chlorin derivatives [179]. For instance, synthetic carbonyl-containing chlorin and boron-based PSs have shown dual applications in PDT and boron neutron capture therapy, a tumor specific radiotherapy, in F98 rat glioma-bearing models [180]. Intriguingly, several studies have designed PS-loaded nanocarriers with the ability to deliver exogenous oxygen to tumor tissue to overcome tumor hypoxic conditions. Among these are perfluorocarbon-loaded NPs, oxygen-encapsulating nanobubbles, and nanosystems with the ability to convert hydrogen peroxide to oxygen [167,181,182]. In another vein, studies have compounded on PDT-induced hypoxia by co-loading PSs with hypoxia-activated prodrugs into NPs to simultaneously induce photodynamic therapy and deliver chemotherapy [183]. Numerous other groups are working towards designing nanoplatforms that can encapsulate multiple cargos to deliver concomitant PDT/chemotherapy [165,170] or immune-photodynamic therapies targeting gliomas [184]. While preliminary results are promising, studies have shown that nanosystem-mediated PDT still exhibits poor drug targeting, premature release into circulation, and lack of real-time drug monitoring [169]. Further work is required before advancing to Phase 0 studies.

## 4. Optimizing Light Delivery

The extent of the PDT therapeutic effect is dependent on the delivery of light. At the optimal excitation wavelength of first-generation PSs (630 nm), light can penetrate tissue to a depth of between 0.8–1.0 cm, with subsequent necrosis expanding 2–7 mm from the point of maximum intensity. While dependent on tissue type and protocol, light dosimetry can be improved by altering light delivery geometry (planar, spherical, cylindrical), light wavelength (longer wavelengths penetrate deeper tissue), light localization (spot delivery vs. interstitial delivery), and light delivery timing (continuous vs. fractionated illumination) [102]. In earlier studies, light was superficially administered by illuminating the margins of the resection cavity or directly onto the tumor tissue surface using laser sources or conventional lamps. To penetrate deeper tissue, later studies have moved toward interstitial (iPDT) or cavitary photoirradiation methods. Interstitial photo-illumination includes the insertion of single or multiple optical fibers into the resection cavity [94]. This type of photoirradiation can be completed following tumor de-bulking or as adjuvant therapy alone. Cavitary photo-illumination occurs following maximum safe tumor resection and includes using a light-diffusing medium or inflatable balloon to evenly disperse the light dose. Geometrically, cavitary photoirradiation covers a larger surface area than interstitially inserted fibers [97]. Several novel devices for cavitary PDT have been developed, including a balloon-based device that is currently being tested in the INDYGO trial, a phase I trial in recruitment in Lille, France [185]. One of the most interesting devices, however, has been an inflatable indwelling balloon catheter developed by Madsen et al. that allows for post-operative photo illumination. The device consists of a balloon applicator and a two-lumen catheter with a self-sealing penetrable membrane. Following surgical insertion and wound healing, the skin can be punctured with a needle mandrill, the balloon expanded with a diffusion medium, and threaded with an optical fiber through the lumen to photo-irradiate the resection cavity. The balloon can later be deflated and removed after the completion of treatment [186].

In general, conventional laser sources are expensive and carry the risk of unwanted tissue heating. As such, more recent studies have moved toward utilizing light-emitting diode (LED) technologies. LEDs have a broad emission spectrum (630–940 nm) and, therefore, have the potential to penetrate deeper brain tissue at lower light energies [65]. LED equipment is smaller, easier to use, inexpensive, and provides a wider irradiation area, in comparison to lasers. In addition to the aforementioned clinical trial combining LEDs with both HpD and BPD [65], 5-ALA PDT studies have shown that blue LEDs more effectively decrease human glioma cell line viability when compared to conventional red LEDs [187].

Delivering a therapeutic light dose to the targeted tissue remains one of the main clinical challenges of PDT. In 1993, Origitano and colleagues were the first to develop an image-based, computer-assisted treatment planning protocol that individualized light dose volume and geometry with promising results [90]. Since then, many other groups have used mathematical modeling, most commonly Monte Carlo simulations, to predict light propagation and absorption within tissue. This helps to ensure adequate light delivery while limiting off-target heating effects [188]. In fact, a pilot trial utilizing theoretical models and 3-D treatment planning to establish patient-specific irradiation schemes has exhibited one of the highest median OS out of all 5-ALA-based clinical trials [114].

For 5-ALA-based PDT in particular, fractionated light delivery, rather than continuous illumination, has proven to be more efficacious at inducing a phototoxic response. Fractionation is induced when light irradiation alternates between incremental “light” and “dark” periods, rather than continuous illumination of the treatment field. These transient periods of light interruption allow for tissue re-oxygenation, which enhances PDT efficacy by (i) maximizing the phototoxic effect of the subsequent light period, (ii) allowing re-localization of the PS to tumor areas following PDT-induced PS photobleaching, and (iii) promoting reperfusion injury, a tissue-damaging mechanism that results from re-vascularization of ischemic tissue [189,190]. Since HpD has been shown to accumulate in cutaneous tissues, often for long periods of time, a fractionated treatment schema is not compatible with HpD or Photofrin-mediated PDT due to toxicity [186]. Over the last decade, a series of in vitro and in vivo pre-clinical studies have been performed comparing 2-fraction, 5-fraction, and continuous light at high and low power for 5-ALA-based PDT. A 5-fraction treatment schema delivered at low power (5 mW) was shown to induce degrees of apoptosis and a peripheral pro-vascular effect with limited necrosis higher than those produced by other schemas [190,191,192,193]. The authors also noted that rats with larger tumor volumes and were treated at higher fluence rates more frequently exhibited fatal intracranial pressures. This condition may be a contraindication of PDT in future studies. While the reported effects of fractionated light delivery show promise, further studies are needed to account for the low diversity in single cell-line-derived xenograft models. Single cell-line-derived tumor models create homogeneous and hypervascularized tumors that do not exhibit spontaneous tumor necrosis and infiltrative patterns characteristic of GBM.Automation of mPDT has also been explored via telemetric device development in rat GBM models. This implant is placed subcutaneously and contains dual functionality for light delivery and light fluence rate monitoring via a tetherless inductive link. While further work is needed to improve device design and therapeutic administration, the reported model demonstrated successful functionality during a 2-week implantation period without serious biological complications [194].

## 5. PDT in Other CNS Tumors

The use of PDT has also been explored in several other common CNS tumors as well, including meningiomas, pituitary adenomas, and pediatric brain tumors. While these studies have surpassed pre-clinical testing to include clinical application, the bulk of reported results are from in vitro testing.

The utility of PDT in meningioma has been explored using both first- [86,87,97,103,195,196,197,198,199] and second-generation [200,201,202,203,204,205,206,207,208] PSs under both in vitro and clinical conditions. The majority of clinical studies in malignant meningioma (MM) have been conducted using first-generation PSs. However, clinical cohort sizes are not large enough to determine conclusive outcomes of therapeutic efficacy. Similar to malignant glioma studies, PS use quickly shifted to second-generation drugs following their development due to their reduced phototoxicity and improved selectivity. Studies using 5-ALA-induced PDT have found that meningiomas demonstrate irregular fluorescent distribution across the tumor, which implicates higher variability in PS metabolism, as compared to malignant glioma. In addition to variable PpIX fluorescence, meningioma cells show lower fluorescence intensity, and therefore PDT efficacy; an attribute that has been remedied by the co-delivery of drugs such as gefitinib (anti-cancer) [204] and ciprofloxacin (antibiotic) [205]. In TS PDT, pre-clinical studies demonstrated two PDT-induced morphologies, both of which mediate tumoricidal effects: apoptotic presentation as characterized by cell body shrinkage and cell necrosis via swelling of the cell body. Additional evidence of cell necrosis was identified at high TS dosages by increased levels of lactate dehydrogenase leakage, a biochemical marker for cell necrosis [206]. Yet, like 5-ALA, TS PDT efficacy in MM is still below that of malignant glioma, warranting further investigation to improve the anti-tumor effect [206]. Continued development of third-generation PSs may also further improve PDT utility in malignant meningiomas.

The merit of PDT is particularly notable in cases of highly invasive or diffuse tumors with ambiguous margins. The potential use of PDT for the treatment of native pituitary adenomas (PAs) presents a very intriguing option as complete resection of these tumors is often unachievable. Pre-clinical studies of first-generation PS-guided PDT in PAs reported successful cytotoxicity and anti-tumor effect in both in vitro and mouse models [209,210]. Currently, clinical studies on PDT use for PAs have used first-generation PSs for therapy. These reports demonstrated high feasibility for PDT use, yielding an approximately 50% increase in tumor tissue retention than gliomas [211]. Additionally, longitudinal monitoring of this cohort found that the majority of patients recovered partial/full acuity of their visual fields [212]. In vitro studies have also probed 5-ALA PDT efficacy across various dosages using immortalized rat pituitary adenoma cells (GH3), AtT-20 cell lines, and human pituitary adenoma cells, quantifying PpIX fluorescence and probing cytotoxicity within surviving cells. These two studies yielded conflicting results which may be attributed to different culture conditions and inconsistent protocols. However, it can be said that all cell lines displayed a 5-ALA-induced, cell line-dependent endogenous fluorescence, suggesting varying PS uptake and metabolism [213], and that dose-dependent toxicity unique to each cell line was evident as well [214]. Further work will be required to better understand the feasibility and efficacy of second-generation PSs before proceeding to in vivo and clinical studies.

Another class of CNS tumors that are generally restrained to subtotal surgical resection and adjuvant treatment are chordomas: a rare, aggressive sarcoma of the skull base and sacrum. Not only do post-resection margins facilitate high rates of infiltrative recurrence, but these tumors are also typically both chemo- and radiotherapy resistant, stipulating alternative anti-tumor treatment modalities. While PDT for chordomas has not been very well studied, two consecutive in vitro cytotoxicity studies have demonstrated an tumoricidal effect following 5-ALA PDT [215], with elevated efficacy following the administration of the antibiotic ciprofloxacin [216]. Given the similar pattern of antibiotic-elevated PDT performance as is found in gliomas, it is likely that, through further study, PDT may be highly efficacious for chondromas, holding promise for future clinical chondroma management.

Although all of the aforementioned studies on PDT efficacy have been performed on adult CNS tumors, pre-clinical and clinical studies have also been performed evaluating the performance of PDT in pediatric brain tumors via second-generation PSs such as 5-ALA and TS [217,218,219]. One of the most recent reports evaluating PDT feasibility for malignant pediatric brain tumors has reported on treatment using TS in children and young adolescents with brain tumors, including diffuse midline glioma (DMG), glioblastoma, and high-grade glioma. Not only was TS PDT found to be safe for use at dosages comparable to those used in adults, but adverse events commonly identified in adult patients were also not found in children following therapy [220]. These results are incredibly encouraging for PDT usage for pediatric CNS tumors. Further reports, such as those from current clinical trials (UMIN000030883 and NCT04738162), however, are required to better define the efficacy of PDT in this application.

## 6. Limitations

### 6.1. Limitations of PDT and Its Synergistic Agents

The application of PDT for the treatment of gliomas is not without limitations. Technical limitations include relatively high costs, the requirement of specialized equipment, multiport lasers, and medical device class III approved light applicators, which are limited to few neurosurgical centers [221]. Biological limitations include variable PS accumulation in tissues, inadequate penetration of light into deeper regions, heterogeneity of response from variant light penetration depth, reduced efficacy in hypoxic regions, and photobleaching [110]. Furthermore, PS uptake is highly dependent on the cell metabolic state. For example, in vitro studies have demonstrated an enhanced PpIX accumulation in the tumor tissues and a complementary increase in the efficacy of PDT following the upregulation or presence of ATP-binding cascade (ABC) transporters inhibitors [222], iron chelators [223], calcitriol [224], arsenic trioxide [225], or NF-kappaB inhibitors [226]. Despite this high relevance to malignant glioma states, further investigation, as is currently being conducted in third-generation PS testing, to improve tumor specificity through alternative avenues, such as protein markers, is warranted.

### 6.2. PDT Efficacy Negatively Influenced by the Harsh Glioma Microenvironment

Largely, there are two limitations to PDT efficacy that occur as a function of the tumor microenvironment (TME). The highly hypoxic nature of the TME reduces oxygen availability for subsequent singlet oxygen production, thereby reducing the success of PDT [167,181]. This limitation must be taken into account during the literature review, as most reports investigating PDT do so in cell lines at atmospheric oxygen concentrations (~20%) as compared to the hypoxic concentrations typical of glioma (~5–15%) [167,227]. This is especially relevant for 5-ALA-driven PDT, which requires approximately 20% more irradiation under hypoxic conditions to invoke equitable rates of therapeutic success as seen under normoxic environments [227].

The second limitation is the competitive uptake of PS by surrounding glial cells and neurons. Competitive uptake reduces PS availability, thereby decreasing photosensitizer accumulation in tumor cells. Successful efforts, however, have modulated local temperature and demonstrated increased PS uptake solely in cancer cells. Studies from as early as 1995 have shown that induction of moderate hyperthermia or hypothermia increases the efficacy of PDT in normal brain [228] and cancer cells. Moderate hypothermia (32–34 °C) has been shown to increase PpIX concentrations in glioma cell lines, in addition to conferring neuroprotection and increasing the PDT therapeutic index [229]. Interestingly, moderate hypothermia has also been demonstrated to increase synergism between PDT and concurrent adjuvant therapies such as conventional chemotherapy [230] or targeted inhibitors [231].

Other studies have also utilized innovative strategies to combat these limitations in a joint fashion. Micro-optical devices and nanoparticle lipid emulsions have been designed to co-deliver photosensitizer and exogenous oxygen to increase local oxygen concentrations [167,232]. These lipid emulsions and nanoparticles also improve the cellular uptake of various photosensitizers, including Ce6 and 5-ALA, and have been shown to lead to significant increases in PDT efficacy and survival rates in glioma-bearing mice [167,181,182].

### 6.3. Innate PDT Resistance

While one of the main advantages of PDT is its ability to circumvent chemotherapeutic resistance, there are certain pathways of resistance to PDT that are either found in or acquired by glioma. This is primarily facilitated by nitric oxide synthase (NOS) and its inducible counterpart (iNOS), which produce intracellular nitric oxide [233,234]. In fact, exposure to 5-ALA PDT leads to an increase in cytoprotective iNOS expression, resulting in a change in phenotype to iNOS/NO-dependent proliferation, migration, and invasion rate [235]. Further study of this pathway revealed that iNOS expression is regulated by acetylation of NF-κB by bromodomain and extra-terminal (BET) protein. However, NO scavengers, iNOS inhibitors, and BET inhibitors can be used to inhibit iNOS/NO-dependent tumor progression while also increasing the rate of apoptosis and PDT therapeutic efficacy [235,236].

While nitric oxide synthesis is one of the more well-studied mechanisms of PDT resistance, there is another mechanism of PDT resistance that has been characterized in gliomas. Studies in U87 cells show that TP53 upregulates ALKBH2 in cells that survive PDT by binding to its promoter, contributing to the reversal of DNA damage, thereby promoting tumor proliferation [237].

### 6.4. Peri-Tumor Edema Limits PDT Efficacy

While PDT can act synergistically with chemo- and immunotherapies, it also increases the extent of peritumoral edema, which, in turn, decreases the therapeutic efficacy of PDT. To address this, researchers have investigated using different loop diuretics, such as torasemide [238] and bumetanide [239], to relieve peritumoral edema. In rats and xenograft models, this has been shown to supplement PDT efficacy, inhibiting tumor growth and prolonging survival.

### 6.5. PDT Drug Interactions and Synergistic Agents

Other limitations of PDT include dose-dependence and unwanted drug reactions. Studies have shown that the effects of PDT can be dose-dependent in a non-linear fashion for certain photosensitizers, and that exceeding a certain dosage may decrease photodynamic toxicity due to changes in intracellular localization [240]. PDT efficacy can be limited by drugs such as phenytoin, an anti-seizure medication, which reduces the accumulation of 5-ALA in glioma cell lines [241]. As such, there is an imminent need for synergistic agents that increase PDT efficacy. While few in number, there are several FDA-approved drugs found on the market that show promise in this arena. Reports have shown that polyphenol PKC*δ* regulators, such as hypericin and rottlerin, enhance PDT-induced apoptosis of GBM cells. Further, drugs such as atorvastatin have shown reduced PDT-induced functional deficits in rats [242,243]. The need for synergistic agents has also resulted in the development of novel technologies and the identification of targets for gene therapy. Recent reports show that the delivery of micelles with peptide irDG-conjugated photosensitizers results in greater BBB penetration and that upregulation of Cx43, affiliated with poor GBM prognosis [244], improves the efficacy of PDT [168,245].

## 7. Conclusions and Future Directions

While the literature depicts a series of PDT advancements in glioma care beginning in the 1980s, the majority of notable studies have been conducted out of a limited number of institutions. These clinical trials have consisted of small, non-randomized, and histologically diverse patient cohorts, and lack “alternative therapeutic” control cohorts, complicating robust assessment. Furthermore, many photosensitizers (PSs) and light delivery methods each produce different phototoxic outcomes, which are further convoluted by the cell-dependent response. This variability is reflected in the diversity of preclinical glioma PDT studies.

Photodynamic therapy is a promising avenue for malignant glioma management, addressing several current challenges in treatment, such as targeted tumor treatment. However, it is unlikely that single-drug PDT will be sufficient to treat the genetically and phenotypically diverse landscape of glioma. While further studies are required, a multimodal regimen of PDT, in combination with standard radiotherapy and targeted immunotherapies, or engineered delivery modalities, bears promising effects and may elevate the current efficacy of PDT to a higher level.

## Figures and Tables

**Figure 1 cancers-15-03918-f001:**
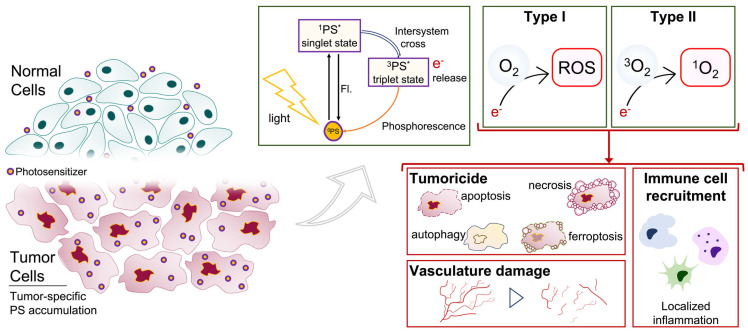
Photodynamic therapy mechanism of action. PDT occurs by light excitation of the tumor-accumulated photosensitizer, which exists at “ground state” (^0^PS). Following excitation, ^0^PS transitions to the singlet, excited state (^1^PS*). As the energy is released, ^1^PS* either drops to the ground state, emitting fluorescence, or drops to the excited triplet state (^3^PS*) through intersystem crossing. Electrons are subsequently released as the triplet state returns to ^0^PS by way of phosphorescence. PDT is executed via two types of photosensitization: Type I, which converts O_2_ to reactive oxygen species (ROS) such as H_2_O_2_, O_2_^−^, or OH^−^; and Type II, which converts ^3^O_2_ to ^1^O_2_. Downstream biological effects on tumor cells include direct tumor cell destruction (apoptosis, necrosis, autophagy, ferroptosis), tumor vasculature damage (microvascular stasis, increased vascular gaps, vessel pruning), and immune cell recruitment via localized inflammation.

**Table 1 cancers-15-03918-t001:** Clinical properties of selected, relevant first- and second-generation photosensitizers.

	Photosensitizer	Intracellular Localization	Excitation Wavelength (nm)	Treatment Window ^a^	Clearance Time	Tumor: Normal Fluorescence Ratio ^b^	Administration	Side Effects
**First Generation**	Porfimer Sodium	Inner mitochondrial membrane	630	48–150 h	4–8 weeks	2.5–4:1	Systemic	Skin sensitization, thrombocytopenia
	Hematoporphyrin derivative [HpD]		408, 510, 630 ^c^	24–48 h	4–6 weeks		Systemic	
	Dihematoporphyrin ether [DHE]		395, 630 ^c^	24–72 h	4–6 weeks		Systemic	
**Second Generation**	5-Aminolevulinic Acid (Levulin^®^, Gliolan^®^)	Early: mitochondria Late: plasma membrane, lysosomes	410, 510, 635 ^c^	4–8 h	2 days	10–20:1	Oral	Skin sensitization, nausea, elevated liver enzymes, anemia
	Talaporfin sodium (Laserphyrin, Aptocine^TM^, LS11, PhotoIon^®^)	Lysosomes	664	12–26 h	15 days	ND	Systemic	Skin sensitization
	Temoporfin [m-THPC; m-tetrahydroxyphenylchlorin] (Foscan^®^, liquid formulation; Foslip^®^, liposomal formulation)	Strong: Golgi apparatus, Endoplasmic reticulum Weak: mitochondria, lysosomes	652	48–110 h	15 days	150:1	Systemic	Skin sensitization
	Boronated protoporphyrin [BOPP]	Lysosomes	630	24 h	4–6 weeks	400:1	Systemic	Skin sensitization, thrombocytopenia
	Benzoporphyrin derivative [BPD]	Lysosomes	680–690	15–30 min.	1–5 days	ND	Systemic	Vascular damage

^a^ Time frame between drug and light administration; ^b^ Tumor tissue: normal tissue fluorescence ratio (T:N); ^c^ Optimal excitation wavelength; ND, not determined.

**Table 2 cancers-15-03918-t002:** Summary of clinical studies using first-generation photosensitizers (PS) for glioma PDT [13,65,80,81,82,83,84,85,86,87,88,89,90,91,92,93,94,95,96,97,98].

	Study Group ^a^ (n, Number of GBM Patients in Study)		Mean Age	PS ^b^	Dose ^c^	Route ^d^	Time Prior to Photoillumination	Photoillumination Method ^e^	Laser/Light Wavelength ^f^ (nm)	Photoillumination Energy (ED unless Otherwise Specified)	Reported Survival ^g^	Survival Statistics		Adverse Events
Perria et al. (1980)	GBM	2	n/a	HpD	5 mg/kg	IV	n/a	n/a	628	720–2400 J/cm^2^	MS	GBM	6.9 mo	n/a
Laws et al. (1981)	rMG	5	14–75	HpD	5 mg/kg	IV	48–72 h	Interstitial	630	30–60 mW/cm^2 §^	TTP	rMG	1–6 mo	Increased skin photosenstivity
McCulloch et al. (1984)	GBM	9	n/a	HpD	5 mg/kg	IV	n/a	n/a	627.8 (>1 laser)	n/a	OS	GBM (*n* = 3)	17–42 mo	Increase in P/O cerebral edema
Muller and Wilson (1985) ^h^	GBM rGBM	1 2	53 32, 44	HpD	2.5 mg/kg 2 mg/kg	IV	24 h	Cavitary (balloon)	630	8–68 J/cm^2^	n/a			None
Kaye et al. (1987) ^i^	GBM rGBM	13 6	45 *** 40 ***	HpD	5 mg/kg	IV	24 h	Interstitial	AI (9) GMVL (14)	70–120 J/cm^2^ 120–230 J/cm^2^	PFS	GBM rGBM	3–13 mo 12–16 wk	No AEs
Kostron et al. (1987) ^j^	GBM	6	63.3	HpD	1.0 mg/cm^3^	IV IA Direct tumor	3 d	LED (*n* = 9) Cavitary (*n* = 5)	620–640 632	422 J/cm^2 §^ <1600 J/cm^2 §^	MS/OS	GBM	12 mo	IA/Direct tolerated without skin phototoxicity
	rGBM (1x)	5	50.8									rGBM (1x)	2–7 mo	
	rGBM (mult)	3	57.0									rGBM (mult)	5 mo	
Muller and Wilson (1987) ^h^	[HpD] GBM	1	52	HpD (8) DHE (24)	2.14 mg/kg 2.08 mg/kg	IV	18–24 h	Cavitary	630	HpD: 32 J/cm^2^ DHE: 23 J/cm^2^	MS	[HpD] GBM	2.9 mo	Skin photosensitivity (*n* = 3)
	[HpD] rGBM	1	32									[HpD] rGBM	5.8 mo	
	[DHE] GBM	7	58.3	Total dose	150 mg							[DHE] GBM	1.1–13.6 mo	
	[DHE] rGBM	7	39.4									[DHE] rGBM	0.2–10.7 mo	
Kostron et al. (1988) ^j^	GBM	8	55 **	HpD	1 mg/cm^3^	IV, IA and/or Direct	3 d	LED Cavitary	590–750 632	422 J/cm^2 §^ 60–200 J/cm^2^	OS	GBM rGBM (1x) rGBM (mult)	0.5–19 m 3–14 mo 1–6 mo	Skin phototoxicity (IA/IV only)
	rGBM (1x)	9												
	rGBM (mult)	3												
	HPD only (n = 9), [HPD+single dose radiation of 4 Gy fast electrons] (n=10), [HPD+single dose radiation+conventional radiotherapy] (n = 4); 3 cases of recurrence and subsequent re-treatment.													
Kostron et al. (1990) ^j^	GBM rGBM	9 18	n/a	HpD	n/a	IV, IA and/or Direct	n/a	Interstitial	630	40–220 J/cm^2^	OS	GBM rGBM	0.5–29 mo 4–13 mo	Increased phototoxicity of the skin
Muller and Wilson (1990) ^h^	GBM rGBM	9 14	48	HpD DHE	5 mg/kg 2 mg/kg	IV	18–24 h	Cavitary	630	24 J/cm^2^	MS	GBM + rGBM	6.3 mo	Increased skin photosensitivity
Powers et al. (1991)	rGBM rMG	1 5	42–61	HPE	2.0 mg/kg	IV	24 h	Interstitial	630	1000 J ^§§^	TTP	rGBM rMG	2–27 wk 6–45 wk	Edema, increased intracranial pressure and skin photosensitivity
Origitano et al. (1993)	rGBM	8	42.2	PNa	2.0 mg/kg	IV	48–72 h	Cavitary Interstitial + post-resection cavitary	630 630	50 J/cm^2^ 100 J/cm per fiber	TTP	rGBM	5–22 mo	Increased skin photosensitivity
Muller and Wilson (1995) ^h^	rGBM	32	41 **	HpD PNa HPE	5 mg/kg 2 mg/kg 2 mg/kg	IV IV IV	12–36 h	Cavitary	630	38 J/cm^2^	MS	[Stratify by light dose] Energy: >1700 J <1700 J	28 wk 29 wk	Edema, increased skin photosensitivity
Popovic et al. (1995) ^i^	GBM rGBM	38 40	n/a	HpD	2.0–2.5 mg/kg	IV	24 h	Cavitary	AI: 1986–1987 GMVL: 1987–1994	240–260 J/cm^2^ (initial pts: 70)	MS	GBM rGBM	24 mo 9 mo	n/a
Muller and Wilson (1997) ^h^	GBM rGBM	11 32	40 58	PNa	2 mg/kg	IV	12–36 h	Cavitary	630	GBM: 30 J/cm^2^ * rGBM: 43 J/cm^2^ *	MS	GBM rGBM	37 wk 30 wk	Increased P/O cerebral edema
Muller et al. (2001) [Phase II] ^h^	rGBM (ED ≤ 50) (ED ≥ 50)	37 (22) (15)	41 **	PNa	2 mg/kg	IV	12–36 h	Cavitary	630	8–110 J/cm^2^	MS	rGBM (ED ≤ 50) (ED ≥ 50)	avg 29 wk (29 wk) (34 wk)	Increased P/O cerebral edema
Muller et al. (2001) [Phase III] ^h^	GBM High light	20	54	PNa	2 mg/kg	IV	12–36 h	Cavitary	630	30–50 J/cm^2^ (low) 110–130 J/cm^2^ (high)	MS	GBM High	92 wk	n/a
	rGBM Low light High light	26 26	48 52									rGBM Low High	29 wk 51 wk	
Schmidt et al. (2004)	Recurrent brain tumors (include GBM)	NS	n/a	PNa	0.75 mg/kg 1.20 mg/kg 1.60 mg/kg 2.00 mg/kg	IV	18–24 h	Laser/LED + Cavitary balloon	Laser: 630 LED: 20–25	100 J/cm^2^	Not specified			No neurotoxicity
Stylli et al. (2004) ^i^	GBM rGBM	31 27	44 ***	HpD	5 mg/kg	IV	24 h	Cavitary	AI: 1986–1987 GMVL: 1987–1994 KTP: 1994–2000	240 J/cm^2 §§^	MS	GBM/rGBM	24 mo	n/a
Stylli et al. (2005) ^i^	GBM rGBM	31 55	47 * 42 *	HpD	5 mg/kg	IV	24 h	Cavitary	AI: 1986–1987 GMVL: 1987–1994 KTP: 1994–2000	240 J/cm^2 §§^	MS	GBM rGBM	14.3 mo 13.5 mo	Increased cerebral edema (*n* = 3)
Muller et al. (2006) ^h^	GBM rGBM	12 37	59 41	PNa	2 mg/kg	IV	12–36 h	Cavitary (balloon/cont. filling with Intralipid) +/− interstitial	AI KTP	58 J/cm^2 §§§^	MS	GBM rGBM	33 wk 29 wk	Skin photosensitivity
Kaneko (2008)	GBM	26	n/a	HPE	3 mg/kg	IV	2 d	Interstitial	640	180 J/cm	n/a			n/a

^a^ Study group: GBM, newly diagnosed GBM; rGBM, recurrent GBM; rGBM (1x), first recurrence of GBM; rGBM (mult), multiple recurrences of GBM; MG, malignant glioma; rMG, recurrent malignant glioma; ED, energy density. ^b^ Photosensitizer: HpD, Hematoporphyrin derivative; DHE, dihematoporphyrin ether; PNa, porfimer sodium; HPE, Hematoporphyrin ether; NS, not specified. ^c^ Dosage units: mg/kg of body weight, mg/cm^3^ of tumor. ^d^ PS administration route: IV, intravenous administration; IA, intra-arterial administration; Direct, direct tumor injection. ^e^ Photo-illumination Method: I/O, intra-operative; P/O, post-operative. ^f^ Laser/Light wavelength: AI, Argon Ion Dye pumped laser (630 nm); GMVL, Gold Metal Vapour laser (627.8 nm); KTP, Potassium titanyl phosphate pumped dye laser (532 nm transduced to 628 nm via dye module). ^g^ Reported survival: MS, median overall survival; OS, overall survival; TTP, time to progression; mo, months; wk, weeks. ^h^ Longitudinal reporting of St. Michael’s Hospital patient series. ^i^ Longitudinal reporting of the Royal Melbourne Hospital patient series. ^j^ Longitudinal reporting of the University of Innsbruck patient series. n/a: not available, AEs: adverse events. ^§^ Power density. ^§§^ Median total dose. ^§§§^ Mean total dose. * median. ** mean age of entire study. *** median age of entire study.

**Table 3 cancers-15-03918-t003:** Summary of clinical studies using second-generation photosensitizers (PS) for glioma PDT [22,62,65,72,111,112,113,114,115,116,117,118,119,120,121,122,123,124,125,126,127,128].

	Study Group ^a^ (n, Number of Patients with Disease and Treatment)		Mean Age	PS ^b^	Dose ^c^	Route ^d^	Time Prior to Photoillumination	Photoillumination Method ^e^	Laser/Light Wavelength ^f^ (nm)	Photoillumination Energy ^g^ (ED unless otherwise specified)	Reported Survival ^h^	Survival Statistics		Adverse Events
Kostron et al. (1998)	GBM rGBM	2 8	61–72 34–72	mTHPC ^x^	0.15 mg/kg	IV	4 d	Interstitial I/O cavitary	KTP 652	300 mW/cm^2 §^ 20 J/cm^2^	TTP MS	rGBM rGBM	4 mo 6 mo	Phototoxicity
Rosenthall et al. (2001/2003) ^i^	GBM rGBM	7 9	51 *	BOPP	0.25–8.0 mg/kg	IV	24 h	I/O fiber diffuser for focused surface irradiation	630	25 J/cm^2^	MS	GBM rGBM	5 mo 11 mo	n/a
Schmidt et al. (2004)	Recurrent brain tumors (including GBM)	NS	n/a	BPD	0.25 mg/kg	IV	3–6 h	Laser fiber optic catheter/balloon LED balloon	680	1,800 J ^§§^ (at 100 J/cm^2^)	Not specified			No cytotoxic effects
Kostron et al. (2006)	GBM	26	n/a	mTHPC ^x^	0.15 mg/kg	IV	4 d	I/O cavitary I/O fiber diffuser	KTP 652	300 mW/cm^2 §^ 20 J/cm^2^	MS	GBM	15 mo	Increased skin sensitivity
Beck et al. (2007)	rGBM	10	51.7	5-ALA	20 mg/kg	Oral	1 h	I/O fiber diffuser for focused surface irradiation	633	100 J/cm^2^	MS		15 m	n/a
Elijamel et al. (2007)	GBM, PDT(+) GBM, PDT(−)	13 14	59.6 60.1	5-ALA --	20 mg/kg --	Oral --	3 h --	P/O cavitary balloon	630	100 J/cm^2^	PFS MS	GBM	8.6 mo 52.8 wk	Deep venous thrombosis (*n* = 2)
Stepp et al. (2005)	GBM	5	n/a	5-ALA	20 mg/kg	Oral	3 h	I/O fiber diffuser for focused surface irradiation	633	100–200 J/cm^2^	n/a			n/a
Stepp et al. (2007)	GBM (a) GBM (b) GBM (c)	5 8 7	n/a	5-ALA	20 mg/kg	Oral	3 h	I/O fiber diffuser for focused surface irradiation	633	(a) 100 J/cm^2^ (b) 150 J/cm^2^ (c) 200 J/cm^2^	n/a			No AEs
Akimoto et al. (2012)	GBM rGBM	6 8	49–82 41–61	TS	40 mg/m^2^	IV	24 h	I/O fiber diffuser for focused surface irradiation (1.0 cm diameter)	664	27 J/cm^2^	PFS	GBM	24.8 mo	Increased photosensivity
Lyons et al. (2012)	Total (GBM) PDT(+) PDT(−) PDT(+): [a], [b] PDT(-): [c], [d]	73 30 43 17, 13 18, 25	59 **	5-ALA	20 mg/kg	Oral	3 h	[a] IORT, I/O cavitary, MSR [b] I/O cavitary, MSR [c] IORT, MSR [d] MSR only	630	100 J/cm^2^	PFS MS	[a] [b] PDT+ PDT-	79 wk 39.7 wk 62.9 wk 20.6 wk	n/a
Johansson et al. (2013)	GBM rGBM	1 4	42 56	5-ALA	20–30 mg/kg	Oral	5–8 h	Interstitial	635	720 J/cm^2^	TTP		3–36 mo	n/a
Muragaki et al. (2013)	GBM	13	47.1 **	TS	40 mg/m^2^	IV	22–27 h	I/O fiber diffuser for focused surface irradiation (1.5 cm diameter)	664	27 J/cm^2^	PFS MS		12 mo 27.9 mo	Increased photosensitivity
Schwartz et al. (2015)	GBM	15	n/a	5-ALA	20 mg/kg 30 mg/kg	Oral	n/a	Interstitial	633	12.96 J ^§§^	PFS MS		16 m 34 m	Transient aphasia, pulmonary embolism
Vanaclocha et al. (2015)	GBM	20	49 ***	DHE mTHPC ^x^	2 mg/kg 0.15 mg/kg	IV	48 h 96 h	I/O cavitary	630 652	75 J/cm^2^ 20 J/cm^2^	PFS MS MS (from 1st diagnosis)		10 mo 9 mo 17 mo	Skin photosensitivity dermatitis
Nitta et al. (2018)	GBM	11	54	TS	40 mg/m^2^	IV	22–26 h	I/O fiber diffuser for focused surface irradiation (1.5 cm diameter)	664	27 J/cm^2^	PFS MS		19.6 mo27.5 mo	Asymptomatic transient peripheral edema
Shimizu et al. (2018)	GBM rGBM	7 7	45–74 40–69	TS	40 mg/m^2^	IV	22–26 h	I/O fiber diffuser for focused surface irradiation (1.5 cm diameter)	664	100 J/cm^2^	n/a			Pulmonary embolism (if vessels are not shielded)
Lietke et al. (2021)	rGBM	37	49.4 *	5-ALA	20 mg/kg	Oral	3–5 h	Interstitial	635	8883 J ^§§^	TTP MS (from 1st diagnosis)	Study combines GBM and AA	7.1 mo 39.7 mo	Transient worsening of pre-existing neurological deficits
Vermandel et al. (2021)	GBM	10	57.1 *	5-ALA	20 mg/kg	Oral	6 h	I/O cavitary	635	200 J/cm^2^	PFS MS		17.1 mo 23.1 mo	No AEs
Kobayashi et al. (2022)	GBM	43	46.7 **	TS	40 mg/m^2^	IV	22–26 h	I/O fiber diffuser for surface irradiation (1.5 cm diameter)	664	27 J/cm^2^	PFS MS		6.3 mo 15.4 mo	No AEs
Kozlikina et al. (2022)	GBM	CR	29	5-ALA + Ce6	20 mg/kg 1 mg/kg	Oral IV	4–4.5 h 3–3.5 h	I/O fiber	660	60 J/cm^2 §§§^	n/a			n/a

^a^ Study group: GBM, newly diagnosed GBM; rGBM, recurrent GBM; rGBM (1x), first recurrence of GBM; rGBM (mult), multiple recurrences of GBM; NS, not specified. ^b^ Photosensitizer: DHE, porfimer sodium; 5-ALA, 5-aminolevulinic acid; TS, Talaporfin sodium; mTHPC, Temoporfin, BOPP, boronated porphyrin; BPD, benzoporphyrin derivative; Ce6, chlorin e6. Approval: unless otherwise noted (^x^), approved for worldwide use, ^x^ EU approval only. ^c^ Dosage units: mg/kg of body weight, μg/g of tumor, mg/cm^3^ of tumor, mg/m^2^ of body surface area. ^d^ PS administration route: IV, intravenous administration; IA, intra-arterial administration; Direct, direct tumor injection. ^e^ Photoillumination Method: iPDT, interstitial PDT; I/O, intra-operative; P/O, post-operative; IORT, intraoperative radiotherapy; MSR, maximum safe resection. ^f^ Laser/Light wavelength: KTP, Potassium titanyl phosphate pumped dye laser. ^g^ ED, energy density. ^h^ Reported Survival: MS, median overall survival; PFS, progression free survival (median); TTP, time to progression (median). ^i^ Survival data reported in 2003. n/a: not available, AEs: adverse events, CR: case report. ^§^ Power density. ^§§^ Median dose. ^§§§^ Total dose. * median. ** mean age of entire study. *** median age of entire study. In contrast to phasic and single-shot (25 mW/cm^2^) photoirradiation, repeated illumination over long durations (weeks) at low fluence rates (≤5 mW/cm^2^) has seen reduced glioma growth [133]. This finding has initiated a new type of PDT called metronomic photodynamic therapy (mPDT), which involves administering light at subthreshold fluences over extended periods of time. mPDT using LED-coupled fiber light delivered continuously over 24 h has increased survival and inhibited tumor re-growth in astrocytoma-bearing rats [134]. Other studies have also shown, under both in vitro and in vivo conditions, that low fluence, long duration organic LED-based PDT bears a significant anti-tumor effect [133,135]. Currently, the only Phase III PDT clinical trial for brain tumors has implemented mPDT, combining 5-day repetitive 5-ALA/Photofrin mPDT with FGS, yielding outcomes that are comparable to current standards of care [115].

**Table 4 cancers-15-03918-t004:** Ongoing PDT clinical trials recruiting for malignant glioma patients.

Study Name	Trial Phase (Study ID)	Type of Cancer	Drug	Principal Investigator
Photodynamic Therapy (PDT) for malignant brain tumor in children	Phase I/II (UMIN000030883)	Brain Tumor (Pediatric)	TS (Leserphyrin)	Kawamata Takakazu (Tokyo Women’s Medical University)
Clinical Safety Study on 5-Aminolevulinic Acid (5-ALA) in Children and Adolescents With Supratentorial Brain Tumors	Phase II (NCT04738162)	Brain Tumor (Pediatric)	5-ALA (Gliolan)	Walter Stummer (Univ. Hospital, Münster)
Stereotactical Photodynamic Therapy With 5-aminolevulinic Acid (Gliolan^®^) in Recurrent Glioblastoma	Phase II (NCT04469699)	GBM	5-ALA (Gliolan)	Walter Stummer (Univ. Hospital, Münster)
PD L 506 for Stereotactic Interstitial Photodynamic Therapy of Newly Diagnosed Supratentorial IDH Wild-type Glioblastoma	Phase II (NCT03897491)	GBM	5-ALA (PD L 506)	Niklas Thon (Univ. Hospital, Munich)
Dose Finding for Intraoperative Photodynamic Therapy of Glioblastoma	Phase II (NCT04391062)	GBM	5-ALA (Gliolan)	Nicholas Reyns (Univ. Hospital, Lille)
Study to Evaluate 5-ALA Combined With CV01 Delivery of Ultrasound in Recurrent High Grade Glioma	Phase I (NCT05362409)	High Grade Glioma	5-ALA (Gliolan)	Alpheus Medical (Wash. Univ. St. Louis, Dent Institute, Northwell Health)

Note: The above table (Table 4) includes information on clinical trials as of 10 January 2023. Data were gathered by searching the National Institutes of Health’s (NIH) clinical trials database (https://clinicaltrials.gov/, accessed on 10 January 2023) as well as the World Health Organization’s (WHO) International Clinical Trials Registry Platform (ICTRP; https://trialsearch.who.int/, accessed on 10 January 2023). Search criteria included “photodynamic therapy” for “glioma” and/or “brain tumor” patients.

## Data Availability

No new data was created or analyzed for this study. Data sharing is not applicable to this article.

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
