# Peer review of "Systematic Review of Photodynamic Therapy in Gliomas"

_cancers, 2023, doi:10.3390/cancers15153918_

Round 1

Reviewer 1 Report

The “Systematic review of photodynamic therapy in gliomas” by Hsi et al provides an overview of the several generations/types of Photosensitizers (PSs) used for photodynamic therapy (PDT), summarises the data from many clinical trials of PSs for glioma therapy, discusses the mechanisms of action of PSs, briefly discusses possible clinical utility of PSs for therapy of other types of central nervous system (CNS) tumours, as well as limitations of their use.

I recommend to accept the review for publication subject to minor rectifications.

Please find below my comments and suggestions.

I recommend to increase text size in the graphical abstract (and all figures throughout the text), to make it more “readable”

Line 66. Here, the authors mention temozolomide for the first time in the text. Thus, they should use its full name, and introduce an abbraviation in the brackets (the way they did it in the line 103). Next, they should use abbreviation only, and the same applies to all abbreviations in the text (PDT, and others)

I also suggest explaining in the text briefly that TM is a drug commonly used for glioma treatment.

Line 135. Do the authors want to elaborate on describing possible molecular mechanisms of PDT-induced blood-brain barrier disruption?

What about synergy of the PDT and PDL-1/PD-1 -directed therapies?

What about PDT for ferroptosis in glioma treatment?

Author Response

Reviewer 1

The “Systematic review of photodynamic therapy in gliomas” by Hsia et al provides an overview of the several generations/types of Photosensitizers (PSs) used for photodynamic therapy (PDT), summarises the data from many clinical trials of PSs for glioma therapy, discusses the mechanisms of action of PSs, briefly discusses possible clinical utility of PSs for therapy of other types of central nervous system (CNS) tumours, as well as limitations of their use.

We thank the reviewer for their thorough review of our manuscript. We have now addressed the reviewer’s comments in the manuscript (edits highlighted) and have discussed below the specific edits with respect to the reviewer’s comments.

I recommend to accept the review for publication subject to minor rectifications.

Please find below my comments and suggestions.

I recommend to increase text size in the graphical abstract (and all figures throughout the text), to make it more “readable”

We thank the reviewer for their comment. We have now adjusted the text size in the graphical abstract (page 2, line 47) as well as all other figures (Figure 1: page 4, line 90) in the text to improve readability.

Line 66. Here, the authors mention temozolomide for the first time in the text. Thus, they should use its full name, and introduce an abbraviation in the brackets (the way they did it in the line 103). Next, they should use abbreviation only, and the same applies to all abbreviations in the text (PDT, and others)

I also suggest explaining in the text briefly that TM is a drug commonly used for glioma treatment.

We thank the reviewer for this point. We have made sure that all other abbreviations are defined prior to usage. Earlier in the introduction, we have previously mentioned the use of adjuvant temozolomide as the current standard of care in lines 53-54 (page 2).

Line 135. Do the authors want to elaborate on describing possible molecular mechanisms of PDT-induced blood-brain barrier disruption?

We thank the reviewer for making this suggestion. We agree that discussing mechanisms of action is beneficial. We have now added a description of PDT-induced blood-brain barrier disruption (lines 132-133) and have also included an additional, more recent citation (published in 2017) which explores PDT-induced BBB disruption.

What about synergy of the PDT and PDL-1/PD-1 -directed therapies?

We thank the reviewer for bringing up this point. We agree that this is an important discussion to add and have now mentioned and provided references to PDT - anti-PD-L1 combination therapies under Section 3.2.1: Combining second-generation PDT with standard therapies.

What about PDT for ferroptosis in glioma treatment?

We thank the reviewer for their important suggestion. The addition of this point provides an additional aspect that reflects the growing literature on ferroptosis. We have now added this modality of cell death in Figure 1 and mentioned it under Section 2.1: PDT induces cell death following PS uptake, accumulation, and activation.

Reviewer 2 Report

      In this research, the authors reviewed the status of photodynamic therapy in gliomas. In my opinion, the current version of this manuscript fits the scope of Cancers and could be accepted after major revision.

My specific comments are in detail listed below:

1.     The graphic figure is of poor quality. The figures still could be improved if possible.

2.     In the part 6 limitation, the authors should better reviewed the acquired immune resistance faced by photodynamic therapy in gliomas and some related tumors, especially the possible increased PD-L1 expression. Some references should be added to this part including 10.1002/adma.202206121.

3.     Some minor mistakes exist in the references. Besides, some references are out of date. The authors should correct it.

4.     In the introduction part, the merits of PDT should be better introduced especially compared with other tumor therapies such as chemotherapy and radiotherapy. 

5.     The clinical transformation barriers of photodynamic therapy in gliomas should be better out-looked.

6.     In Table 2, some too detailed information was given. It’s better to summary it.   

Author Response

Reviewer 2

 In this research, the authors reviewed the status of photodynamic therapy in gliomas. In my opinion, the current version of this manuscript fits the scope of Cancers and could be accepted after major revision.

We thank the reviewer for their evaluation of our manuscript and their feedback which has helped elevate the quality of this paper. We have addressed the reviewers comments in the manuscript (highlighted) and have described edits in detail below.

My specific comments are in detail listed below:

  1. The graphic figure is of poor quality. The figures still could be improved if possible.

We thank the reviewer for this feedback. We have now re-extracted the figures at 330 dpi and have inserted them into the manuscript.

  1. In the part 6 limitation, the authors should better reviewed the acquired immune resistance faced by photodynamic therapy in gliomas and some related tumors, especially the possible increased PD-L1 expression. Some references should be added to this part including 10.1002/adma.202206121.

We thank the reviewer for this comment. While most studies discuss a positive correlation between PDT and immune response, there still remains a possibility of decreased immune infiltration of the tumor following PDT. We have discussed this point as well as the synergistic effects of PDT and anti-PD-L1 immunotherapy in Section 3.2.1. The reviewer’s suggested citation and additional references to several reviews which provide detailed mechanisms of PDT-driven immune cell recruitment has been added to the manuscript.

  1. Some minor mistakes exist in the references. Besides, some references are out of date. The authors should correct it.

We thank the reviewer for taking note of this. The references have now all been reformatted to fit Cancers/MDPI reference requirements. In an effort to provide a comprehensive review of all PDT use in gliomas from its initial introduction, we have conducted review of early clinical trials (1980). For review of experimental and mechanistic information, however, we have endeavored to provide up-to-date references and a large majority of the references included in this manuscript are from the last 5-10 years.

  1. In the introduction part, the merits of PDT should be better introduced especially compared with other tumor therapies such as chemotherapy and radiotherapy. 

We thank the reviewer for this suggestion. We have now included brief discussion of the benefits of PDT over standard therapies in the introduction.

  1. The clinical transformation barriers of photodynamic therapy in gliomas should be better out-looked.

We thank the reviewer for this comment. In the limitations section we have discussed several hurdles that need to be overcome in PDT treatment in the context of gliomas. We have also provided review of supplemental drugs, formulations, and methods that help overcome these barriers.

  1. In Table 2, some too detailed information was given. It’s better to summary it. 

We thank the reviewer for this feedback. We agree that Table 2 does contain extensive information, in particular, with respect to the breakdown of patient cohorts, i.e. GBM and single/multiple recurrent GBM. This was done intentionally since the morphology and characteristics of glioblastoma and recurrent glioblastoma (single and multiple recurrences) are highly heterogeneous and diverse. In order to capture and summarize the effects of variable PDT modalities within each disease category, we have extensively described each of the studies with as much detail as possible. This type of review has not yet been done previously and we believe that this detailed summary gives novelty to this manuscript.

Round 2

Reviewer 2 Report

The current version of this manuscript could be accepted.